# Antimicrobial Resistance in Livestock: A Serious Threat to Public Health

**DOI:** 10.3390/antibiotics13060551

**Published:** 2024-06-13

**Authors:** Roberto Bava, Fabio Castagna, Carmine Lupia, Giusi Poerio, Giovanna Liguori, Renato Lombardi, Maria Diana Naturale, Caterina Mercuri, Rosa Maria Bulotta, Domenico Britti, Ernesto Palma

**Affiliations:** 1Department of Health Sciences, University of Catanzaro Magna Græcia, 88100 Catanzaro, Italy; roberto.bava@unicz.it (R.B.); studiolupiacarmine@libero.it (C.L.); rosamaria.bulotta@gmail.com (R.M.B.); britti@unicz.it (D.B.); palma@unicz.it (E.P.); 2Mediterranean Ethnobotanical Conservatory, Sersale (CZ), 88054 Catanzaro, Italy; 3ATS Val Padana, Via dei Toscani, 46100 Mantova, Italy; giusi.poerio@gmail.com; 4Local Health Authority, ASL, 71121 Foggia, Italy; giovanna.liguori@aslfg.it; 5IRCCS Casa Sollievo Della Sofferenza, San Giovanni Rotondo (FG), 71013 Foggia, Italy; renato.lombardi@aslfg.it; 6Ministry of Health, Directorate General for Health Programming, 00144 Rome, Italy; mariadiananaturale@gmail.com; 7Department of Experimental and Clinical Medicine, University “Magna Graecia”, 88100 Catanzaro, Italy; c.mercuri@unicz.it; 8Center for Pharmacological Research, Food Safety, High Tech and Health (IRC-FSH), University of Catanzaro Magna Græcia, 88100 Catanzaro, Italy

**Keywords:** antimicrobial resistance, public health, resistance mechanisms, farm animals, environmental residues, One Health approach

## Abstract

Antimicrobial resistance represents an alarming public health problem; its importance is related to the significant clinical implications (increased morbidity, mortality, disease duration, development of comorbidities, and epidemics), as well as its economic effects on the healthcare sector. In fact, therapeutic options are severely limited by the advent and spread of germs resistant to many antibiotics. The situation worldwide is worrying, especially in light of the prevalence of Gram-negative bacteria—*Klebsiella pneumoniae* and *Acinetobacter baumannii*—which are frequently isolated in hospital environments and, more specifically, in intensive care units. The problem is compounded by the ineffective treatment of infections by patients who often self-prescribe therapy. Resistant bacteria also show resistance to the latest generation antibiotics, such as carbapenems. In fact, superbacteria, grouped under the acronym extended-spectrum betalactamase (ESBL), are becoming common. Antibiotic resistance is also found in the livestock sector, with serious repercussions on animal production. In general, this phenomenon affects all members of the biosphere and can only be addressed by adopting a holistic “One Health” approach. In this literature overview, a stock is taken of what has been learned about antibiotic resistance, and suggestions are proposed to stem its advance.

## 1. Introduction

Modern medicine is built around antibiotics. Neonatal mortality has decreased as a result of their use. They also play a crucial role in advanced therapies like chemotherapy and invasive surgical operations. They are a vital resource in the age of robotically guided surgical procedures and for treating infections that develop after illnesses. On the other hand, the likelihood of the fear of incurable infections really coming true is growing because of the prevalence of multidrug-resistant bacterial infections worldwide. The term antimicrobial resistance (AMR) refers to the ability of a bacterium to resist the activity of one or more antibiotics (multidrug resistance, MDR) and to survive and grow in the presence of a concentration of the antibacterial agent that was generally sufficient to inhibit its multiplication or was capable of killing it [1]. The often excessive and incorrect consumption of antibiotics over time has always been accompanied by the onset of resistance in bacteria (bacterial evolution driven by antibiotics [2]), thus creating the need to produce new molecules capable of overcoming this problem. As early as 2017, the World Health Organization (WHO, Geneva, Switzerland) reported that there would be a global shortage of antibiotics due to the fact that the majority of currently prescribed drugs were created by altering pre-existing classes and have short half-lives [3]. As a consequence, AMR is a serious public health emergency that could have a strong economic impact. According to data provided by the European Commission, the annual losses resulting from antimicrobial resistance (AMR) in the European Union (UE) are estimated to be EUR 33,000, with corresponding healthcare expenditures of around EUR 1.5 billion [4]. The first factor contributing to the antimicrobial problem is the overuse of antibiotics in the human, veterinary, agricultural, and livestock sectors. In human medicine, overuse is often accompanied by improper dosing, poor adherence to treatment guidelines, self-prescribing behavior, and, thus, self-medication [5,6]. These behaviors are more common in poor nations with lax legal policies. For instance, India is one of the world’s biggest users of human antibiotics [7], which are frequently misprescribed by general practitioners. Other factors improving AMR, which are not immediately apparent, include food import–export, environmental pollution, and social factors, such as international and intercontinental travel (Figure 1) [8]. In livestock breeding, antibiotics are administered to animals to promote their growth and prevent pathologies [9]. Around 75 percent of antibiotics are not absorbed by the animals and are instead eliminated from the body via urine and feces, which may pollute and damage the environment in which they are expelled [10]. The use of numerous antibiotics at sub-therapeutic doses and for extended periods of time has favored the fixation of genes conferring resistance, which can then be passed on to human pathogens or the intestinal microbiota via contaminated food or the environment [11]. It is common practice in many nations to fertilize agricultural soil with animal feces, which increases the dissemination of resistance genes that are conveyed via food and groundwater [12]. Gene resistance transfer can take place in food, water, and the digestive tracts of both people and animals [13,14]. Antimicrobial resistance genes can be acquired by foods through several means, such as fecal contamination during animal slaughter, which can infect food with resistant bacteria [14]. Cross-contamination of food can also occur when it is handled by consumers, from the environment, or after processing. Certain food products, such as fresh vegetables, fruit, and raw milk, can be consumed raw and may contain live and viable bacteria at the time of consumption. Aquaculture and plant protection are two further fields of use for antibiotics [11,15]. A significant chance of gene transfer may result from all of this. Therefore, through Darwinian selection, bacteria have evolved strong defenses against a variety of harmful compounds [16]. If antibiotic resistance increases, veterinary medicine could suffer major animal welfare issues. The so-called reserve-antimicrobial medicines (ketolids, oxazolidinones, glycylcyclines) will probably only be used in human medicine since using antibiotics on animals might lead to the transmission of resistance to humans [17]. The rise of plasmid-mediated colistin resistance in human, animal, and zoonotic diseases, for instance, highlights the significance of this expanding issue. In addition to being utilized in human medicine, colistin is widely used in veterinary medicine, particularly as a feed additive in many regions of the globe to aid in the development of food animals [18]. The evolution of colistin resistance has been attributed, at least in part, to the industrial-scale use of colistin as a growth booster for food animals [19]. A study by the World Health Organization [20] states that antimicrobial resistance has reached concerning levels in several regions of the globe, indicating that the 21st century may indeed see the emergence of a “post-antibiotic era”. In veterinary practice, the minimum and sensible use of currently available antimicrobial medications should be promoted. Furthermore, tactics including immunization, immunomodulation, and pre- and pro-biotics should be taken into consideration as options for treating bacterial infections [21]. However, putting such strategies into practice is difficult since it calls for an understanding of the epidemiology and evolution of antibiotic resistance [22,23]. Since human, environmental, and animal health are all interdependent, it is imperative that we solve the AMR issue in the modern era by embracing the “One Health” approach. Important facets of antibiotic resistance are reviewed in this manuscript, with particular attention on farm animals. An initial section discussing the mechanisms that confer AMR to bacteria is followed by two sections looking at the relationship between animal bacteria, human bacteria, and livestock breeding; finally, we present two sections considering strategies to stem antibiotic resistance.

## 2. Antimicrobial Resistance Transmission

Chromosomal mutations (chromosomal resistance) or acquisition of genetic material deriving from bacterial populations related or not to the recipient one (transferable resistance) are at the basis of the problem of antibiotic resistance [2]. During the bacterial genome’s replication, spontaneous mutations or alterations take place in the genome, primarily in the chromosomes. In vitro rates of 10^−6^ to 10^−12^ indicate that this is a low-probability occurrence, yet it might be very frequent in the ubiquitous large bacterial population [24]. The spread of chromosomal mutations occurs through vertical transmission and independently of the presence of the antibiotic; generally, these mutations are corrected by cellular mechanisms, and, for such reasons, they are rare. However, any mutation that confers full or partial resistance to a specific antibiotic will be retained and vertically transferred to progeny cells, giving mutated cells an additional advantage over non-mutated (i.e., parental) cells in terms of growth or survival (depending on the concentration of the antibiotic). When target-site gene mutation(s) for a particular antibiotic occur, de novo mutations may result in bacterial resistance to that antibiotic alone or to a class of antibiotics. Fluoroquinolone-resistant isolates are an example of de novo mutations that solely produce resistance to a particular antibiotic or family of antibiotics. Fluoroquinolones target the type II topoisomerases found in bacteria, namely DNA gyrase (*gyr*A and *gyr*B) in Gram-negative bacteria and DNA topoisomerase IV (*par*C and *par*E) in Gram-positive bacteria. Spontaneous mutation of genes (*gyr*A and *gyr*B) coding for DNA-gyrase results in the production of DNA-gyrase that is not inhibited by fluoroquinolones. The substrate structure or DNA of the *gyr*A and *par*C genes is changed by non-synonymous mutations that occur in these genes, primarily in a short DNA region called the quinolone resistance-determining region (QRDR) [25]. Due to these target-site amino acid changes, fluoroquinolone’s binding affinity to mutant *gyra* and *parc* enzymes is decreased [26], thus granting the antibiotic class a unique resistance. Additionally, research has shown that qnr genes mediate quinolone resistance via plasmids. Seven fluoroquinolone resistance genes have been found and published so far—*qnr*A, *qnr*B, *qnr*S, *qnr*C, *qnr*D, and *qnr*VC. Quinolone resistance may also be caused by the presence of *qnr* genes, the efflux pump *qep*A, and the aminoglycoside resistance gene *aac(6})-ib* [27]. Recently, other genes conferring drug resistance in food animals have been discovered, such as the pig’s mobile colistin resistance (*mcr*-1), the chicken and pig’s plasmid-mediated tigecycline resistance gene (*tet*(X)), and β-lactam resistance gene (*blactx*–*m*–8) in chickens [19,28,29]. In addition to mutations conferring exclusive resistance to a particular antibiotic or class of antibiotics, de novo mutations often result in resistance to a broad range of functionally unrelated antibiotics. In most cases, mutations that result in the overexpression of genes producing efflux pump proteins enable bacteria to become multidrug resistant (MDR) [30,31,32]. For example, the resistance-nodulation division (RND) family of efflux pumps, which includes the AcrAB-TolC, may supply MDR to Gram-negative bacteria through de novo mutations occurring in the AcrAB promoter region [33], mutations occurring in regulators of the AcrAB efflux pump *mar*A [34], *sox*S [35], *rob* [36], and *acr*R [37], as well as mutations affecting the repressor *ram*R [38]. In addition to causing MDR, efflux pump-related mutations may also significantly change an individual’s sensitivity to antibiotics, increasing susceptibility to certain classes of antibiotics while imparting resistance to others [31]. For example, the presence of tetracycline efflux genes, such as *tet* (A–E), which are present in a broad range of bacteria, may result in an enhanced clearance of tetracyclines [39]. De novo mutations have been shown to produce both a large down-regulation of the main porins OmpF and OmpC and an overexpression of MDR pumps [32]. As a result, the inflow and efflux of a mutant bacterial cell are decreased and enhanced, respectively. The phenomenon of transferable resistance is more significant and involves the dislocation of bacterial genes capable of conferring resistance; such Gene transfer is mediated by plasmids, integrons, and transposons and can occur between chromosomal and extra-chromosomal DNA within the same bacteria species and otherwise (horizontal transmission), thanks to the processes of transformation, conjugation, and transduction (Figure 2) [40]. These allow the recipient bacteria to acquire the ability to implement one or more resistance mechanisms in order to defend oneself from antibiotic activity. These mechanisms can be classified based on the biochemistry pathway involved and involve the ability to modify the antibiotic molecule, to preclude reaching the target sites, to modify or protect, and to replace or bypass the target sites. According to Berends et al. (2001) [41], conjugation-induced antimicrobial resistance accounts for around 85% of treatment failure in staphylococci (β-lactams), Enterobacteriaceae (ampicillin, sulpha/trimethoprim, gentamicin, and chloramphenicol), and enterococci (vancomycin). The spread of mobile genetic elements, such as plasmids (which can replicate independently), transposons (which can replicate independently but move through transposases), or integrons/gene cassettes (which can replicate independently but move through site-specific recombination) is what makes up the rapid horizontal gene transfer that occurs during conjugation [42,43]. Two frequent and significant vectors for the intra- and inter-species or inter-genus [44] spread of antibiotic resistance are plasmids and conjugative transposons, which are tiny segments of the bacterial chromosome that encode enzymes for their translocation. Antimicrobial resistance genes are often found in a variety of environmental bacterial taxa, such as Vibrionaceae [45], Shewanella [46], and Kluyvera [47]. The chromosomal DNA of the environmental Kluyvera genus has been proposed as the probable source of the *bla*CTX-M genes, which are the most common cause of extended-spectrum β-lactamases (ESBLs) in Enterobacteriaceae globally [47]. More recently, the OXA-48-type carbapenem-hydrolyzing β-lactamase genes that are present in Enterobacteriaceae are thought to have come from the Shewanella genus’s chromosome, which is found in aquatic environments [46]. Different integron types are another efficient means of transferring antimicrobial resistance horizontally across different bacterial species, particularly between Gram-negative bacteria [48]. Five kinds of integrons impart resistance to therapeutically important antibiotics, according to the literature [49]. While classes 2 and 3 are also recovered from clinical isolates, their rates are lower than those of class 1 integrons [48]. Class 1 integrons are the most often detected among clinical isolates. One example of a plasmid with a class I integron, several MRRs, and heavy metal resistances is pIMP4-SEM1, a plasmid that was recently isolated from *S. enterica Typhimurium* and detected in an Australian cat. Critically essential antimicrobials (carbapenems) were among the nine antimicrobial classes to which this IncHI2 plasmid (pIMP4-SEM1) conveyed resistance. A blaIMP-4-qacG-aacA4-catB3 cassette array containing the carbapenem-resistance gene (*blaimp-*4) was linked to a class 1 integron [49]. There are two further kinds of integrons: class 5 integrons are discovered on the pRSV1 plasmid, which was originally derived from *Alivibrio salmonicida*, and class 4 integrons are recovered from the human disease *Vibrio cholera* [48]. These integrons are thought to contain around 130 distinct resistance gene cassettes, and their phylogeny and extreme variety suggest that these genes have been periodically retrieved from diverse genetic backgrounds [50]. Integrons are thought to be a key factor in the global spread of antibiotic resistance [48]. Therefore, the existence of resistance determinants, the vertical (clonal) or horizontal transmission of these resistance determinants, and selection pressure are the three factors that cause the fast development of antimicrobial resistance in a bacterial population [21].

## 3. The Mechanisms of Antimicrobial Resistance

Resistance determinants, which may spread either vertically or horizontally, provide specific methods to bacteria so they can avoid the effects of the antimicrobial drug. Antimicrobial resistance is caused by a variety of mechanisms, such as the active efflux of antibiotics, decreased drug entry into pathogen cells, enzymatic conversion of antimicrobial drugs to inactive products, development of biofilms, modification of drug targets, and defense of antimicrobial targets. Three enzymatic pathways of antibiotic resistance are known to exist: redox reactions, group transfer, and hydrolysis [51]. The oldest and most varied enzymes that break down antibiotics are called β-lactamases; they cleave the β-lactam ring of antibiotics belonging to the penicillin group, making them inactive. The ESKAPE group of pathogens, which includes *Enterococcus*, *Staphylococcus aureus*, *Klebsiella pneumoniae*, *Acinetobacter baumannii*, *Pseudomonas aeruginosa*, and *Escherichia coli*, is known for its widespread β-lactamase-mediated resistance to antibiotics. These infections are often linked to an important economic burden and a higher risk of death [52,53]. β-lactamases that could break down modified penicillins and semisynthetics like ampicillin, amoxicillin, and methicillin gradually emerged as a result of penicillin introduction. TEM-1 was the first plasmid-transferable β-lactamase, and it was followed by SHV-1 and TEM-2 [54,55]. While SHV-1 is less prevalent, TEM-1 is the most common route of resistance to ampicillin, even though both have the same affinity for the antibiotic. However, 60% of the amino acids in the TEM-1 and SHV-1 are identical, and clavulanic acid, tazobactam, and sulbactam block them. The age of synthetic cephalosporins, which were believed to combat β-lactamases, began with the discovery of cephalosporin C at the start of the 1960s. In terms of structure, penicillins have the β-lactam ring joined to a thiazolidine ring into five members, whereas cephalosporins have a fused β-lactam ring with a six-membered dihydrothiazine ring [56]. However, via point mutations, TEM-1 and SHV-1 lactamases gave rise to extended-spectrum β-lactamase (ESBL) enzymes that can hydrolyze a variety of cephalosporins [54]. Clavulanic acid inhibits ESBLs, which hydrolyze a wide range of cephalosporins, including first, second, and third-generation cephalosporins, as well as aztreonam, but not cephamycins or carbapenems [54,57]. Another typical mechanism of resistance to rifampicin, fosfomycin, and macrolides is enzymatic hydrolysis. Numerous Enterobacteriaceae members generate plasmid-encoded *ere*A and *ere*B esterases genes, which hydrolyze the 14- and 15-membrane macrolide macrolactic rings, including azithromycin, clarithromycin, and erythromycin A [58]. The macrolide antibiotic with a changed structure will no longer be able to attach to the ribosome’s preferred target location [59]. The chromosomally encoded enzyme FosX, which is reliant on the manganese ion (Mn^2+^) and utilizes water to fracture the epoxy ring of fosfomycin, is another significant route of enzymatic degradation. FosA, FosB, and the two epoxy kinases FomA and FomB are additional metalloenzymes that alter fosfomycin [60]. Whereas FosB is a Mg^2+^ thiol-S-transferase, FosA is a glutathione-S-transferase that is reliant on Mn^2+^ and K^+^. The process results in the production of an inactive medication by adding glutathione or thiol groups to the oxirane ring of fosfomycin [61]. The oxirane ring of fosfomycin is phosphorylated by the FomA and FomB kinases using ATP and Mn^2+^ ions. Resistance to some antibiotics, such as aminoglycosides, rifamycins, macrolides, epoxides, and chloramphenicol, is conferred by the enzymatic alteration of antibiotics by the transfer of functional groups, such as acyls, glycosyls, ribosyls, nucleotides, phosphoryls, or thiols [62]. The resistance of several aminoglycoside antibiotics is caused by aminoglycoside-modifying enzymes (AMEs), such as Oadenyltransferases (ANTs), O-phosphotransferases (APHs), and N-acetyltransferases (AACs). These enzymes alter the amino or hydroxyl groups of aminoglycosides in a way that prevents them from attaching to their 30S ribosomal targets [63]. Similarly, resistance to rifampicin in Gram-negative bacteria is often caused by a plasmid-encoded ADP-ribosyltransferase (Arr-2) [64]. Similar to this, the enzyme chloramphenicol acetyltransferase (CAT) modifies chloramphenicol by acetyl-CoA-dependently acetylating its 3-hydroxyl group [65]. The ribosome’s 50S subunit, the modified antibiotic’s target location, is not bound by it. The bacterial enzymes listed above have the ability to change drug targets, which in turn prevents drug binding and confers resistance. Peptidoglycan is a well-known example of one of these targets. In the arms race between antibiotic substances and antimicrobial-resistant bacteria, mutations that preserve functionality while decreasing the capacity of an antimicrobial agent to attach to the target site have been a constant. Apart from peptidoglycan, lipopolysaccharides, ribosomes, and nucleic acid enzymes have also been linked to the modification of target sites. Macrolides, lincosamides, and streptogramin B bind to the 50S ribosomal unit [66]. The term “MLS(B)-type resistance” refers to the resistance to these particular antimicrobials that arises from a post-transcriptional change of the 50S ribosomal subunit’s 23S rRNA component, which is implicated in the methylation or dimethylation of important adenine bases in the functional peptidyl transferase site [66,67]. Numerous species, including *Propionibacteria* [68] and *Helicobacter pylori* [69], have been linked to resistance to the macrolide group of antibiotics due to changes in the 23S rRNA, specifically in the vicinity of the methylation site. A change in the 50S subunit’s L4 and L22 proteins has been linked to *Streptococcus pneumoniae* resistance to macrolides [70]. Tetracycline and aminoglycosides target the 30S ribosomal unit and work by inhibiting mRNA decoding [71]. Resistance to this family of antibiotics is conferred by mutations in the gene encoding 16S rRNA [72]. The lack of these mutations in *Mycobacterium* isolates that are sensitive to kanamycin further supports the finding made by Suzuki and colleagues that substitutions at positions 1400, 1401, and 1483 cause kanamycin resistance in clinical isolates of the bacteria [73]. Active drug efflux mechanisms may be involved when intact antimicrobial drugs penetrate bacterial cells and drug targets are easily accessible. All bacteria have efflux pumps, which are essential to their physiology and play a number of roles in the body’s processes, including the removal of hazardous metabolic products and the preservation of homeostasis. Efflux pumps are essential for the survival of clinically significant bacteria, including methicillin-resistant *Staphylococcus aureus*, *Pseudomonas aeruginosa*, *Klebsiella pneumoniae*, and MDR *Mycobacterium TB*. They also aid in the generation of resistant strains of these bacteria. These bacterial multidrug efflux pump systems are driven by two sources of energy: electrochemical ion gradients (or ionic driving forces, known as secondary active transport [74,75]) and ATP hydrolysis (known as primary active transport [76]). To power the drug’s translocation via the transporter outside the membrane, ATP is hydrolyzed from bacterial cells. In this sense, the bacterium acquires drug resistance while the transporter substrate actively builds up outside the cell [76]. The family of ATP-binding cassette (ABC) efflux pumps is one of the most researched main active drug efflux mechanisms in bacteria [77,78]. In bacteria such as *Mycobacterium* TB, *Acinetobacter baumannii*, *Streptococcus pneumoniae*, *Staphylococcus aureus*, etc., ABC group efflux pumps produce bacterial resistance to therapeutically important medicines. Bacterial resistance to several structurally different antimicrobial drugs is also conferred by secondary active transporters [79,80]. Based on similarities in amino acid sequences, architectures, and mechanisms of energization, these active antimicrobial efflux pump systems have been categorized into numerous major superfamilies of related proteins during the last 30 years. The major facilitator superfamily (MFS) [81], the drug/metabolite transporter (DMT) superfamily [82], the multidrug and toxic compound extrusion (MATE) family [83], the proteobacterial antimicrobial compound efflux (PACE) transporter superfamily [84], and the resistance-nodulation-cell division (RND) superfamily [85] are the current superfamilies and MOP [multidrug/oligosaccharidyl-lipid/polysaccharide] transporters [83,86]. The mechanisms of antimicrobial resistance are summarized in Table 1.

## 4. Relationship between Animal and Human Bacteria

Numerous studies have shown that farm animals are a source of ARB (including *Salmonella*, *E. coli*, *K. pneumonia*, *S. aureus*, and others) and ARGs (Table 2) and that intimate contact with farm animals may result in the transmission of ARB to humans [87,88]. Antimicrobial-resistant bacteria often spread to humans via very complicated and unpredictable pathways. There are two main ways to acquire antibiotic resistance: (i) directly through contact with animals that produce food or human carriers; (ii) indirectly through the food chain or through exposure to high-pollution areas that are high in antimicrobial resistance (e.g., hospitals, nosocomial acquisition, manure, waste water, and agriculture land). The high prevalence rate of antimicrobial-resistant bacteria among people who have close contact with animals, particularly farm workers [89,90] and veterinarians [91], has been shown in several studies investigating the transfer of antimicrobial-resistant bacteria from animals to humans. It has been shown that people and animals may exchange AMR bacteria. For example, evidence of closely related *Klebsiella pneumoniae* strains sharing a sequence type (ST, as defined by multi-locus sequence typing, or MLST) was recovered in a genomics study analyzing the correlation of urinary tract infections and meat retail in the same US city [92]. The methicillin resistance determinant was acquired during circulation in animal populations, and this resistant strain was subsequently transferred to humans, according to an analysis of a lineage of methicillin-resistant *Staphylococcus aureus* known as CC398. The progenitor strain of this lineage was methicillin sensitive and circulated in the human population [93]. Although human-to-animal transmissions did happen, further research revealed that the resistant strain was, in fact, more often transferred from animals to people [94]. Levy et al. [95], among the first authors, documented a direct transfer of multidrug-resistant *E. coli* between animals and between animals and people. The *E. coli* strains employed in this study were those that had the R plasmid, which displayed resistance to many antibiotic families, including streptomycin, tetracycline, chloramphenicol, and sulphonamides. Four chickens were infected with multidrug-resistant strains of *E. coli*, and each infected chicken was kept in a cage alongside fifty uninfected chickens. Furthermore, two groups of chickens were given a meal laced with tetracycline, while the other two groups were provided a diet devoid of antibiotics. The multidrug-resistant *E. coli* strain with the test R plasmid was only discovered by the scientists in chickens given feed laced with tetracycline. It is interesting to note that over the course of the experiment, fecal samples from people who either worked or lived on this specific farm contained R plasmid. The significance of the zoonotic transmission pathway in the development of antibiotic-resistant bacteria was well shown by the study’s authors. The significance of using antibiotics as growth promoters in the development of multidrug resistance phenotypes was also brought to light by this research. Meijs et al. (2021) [88] looked at the incidence and occurrence of AmpC β-lactamase (AmpC)-producing *K. pneumoniae* and extended-spectrum β-lactamase (ESBL)-producing *E. coli* in farmworkers and household members in the Netherlands. ESBL-producing *E. coli* and *K. pneumoniae* were found in 9.8% of workers, compared to 5% in the Dutch population, according to PCR and sequencing data. This finding may be connected to close contact with animals. The most common ESBL genes in workers were *bla*_CTX−M−15_ (48.5∼64.3%), *bla*_CTX−M−14_ (7.1∼18.2%), and *bla*_DHA−1_ (12.1%). Furthermore, 13% of household members had the same ESBL gene as workers, and 17.4% of members had *K. pneumonia* and *E. coli* that produce ESBLs. The information suggested that either direct or indirect animal interaction may be a source of antibiotic-resistant bacteria (ARBs) and antibiotic-resistant genes (ARGs) [88]. Vines et al. (2021) [96] used multiplex PCR to examine the dissemination of plasmids linked with *mcr* in *E. coli* after collecting 210 fecal samples from farm animals and the farmer. Eighteen colistin-resistant *E. coli* isolates with thirty-three virulence factors and *mcr*-1 were found; these included thirteen from calves, four from pigs, and one from the farmer. Additionally, the farmer acquired the *mcr*-1 gene from the calf, increasing the possibility that a farm may act as a zoonotic reservoir for ARB [96]. The spread of resistant bacterial forms, in addition to selective pressure, is catalyzed by population density. Close contact between large numbers of food-producing animals can play an important role in the spread of antibiotic-resistant bacteria. A study team [97] looked at enterohemorrhagic *E. coli* ecology. A statistically significant (*p* = 0.003) positive connection was found between the incidence rate of this zoonotic infection and cattle density in many feedlot operations. This problem is even more important on intensive farms, considering that a large portion (70%) of the antibiotics given to animals are excreted in the environment [10]. ARBs and ARGs are stored in the animal gut microbiota, and the primary factor causing the predominance of ARGs is antibiotic usage [98]. Using 81 fecal samples taken from a pig farm, Fournier et al. (2021) [99] found that *Enterobacterales* were resistant to aminoglycoside, β-lactam, and colistin. Two *K. pneumoniae* isolates that were resistant to colistin and one *Enterobacter cloacae* isolate that had a truncated *mgr*B encoding resistance gene to colistin were found, along with thirty-eight β-lactam-resistant *E. coli* isolates and one *E. cloacae* isolate that carried plasmids with *bla*_CTX−M−1_, *sul2*, and *tetA* genes, respectively, conferring resistance to β-lactam, *sul*, and tetracycline. The findings suggested that the likelihood of resistance spreading in the environment may be raised by the presence of β-lactam-resistant *E. coli* [99]. Zhu et al. (2020) [98] used a metagenomic approach to investigate the distribution and frequency of ARGs in 30 fecal samples that they gathered from various cattle farms. The findings indicated that the variety and abundance of ARGs in beef and dairy cattle were comparable to those in individual yaks and were much greater, perhaps indicating a link to long-term antibiotic usage. When opposed to direct acquisition, the process of acquiring antimicrobial-resistant microorganisms indirectly is often more complicated. Antibiotics used in human, animal, and agricultural medicine are known to release a significant quantity of their active forms into the environment [100]. Consequently, a selection pressure resulting from the existence of active antibiotic compounds in the environment might lead to the formation of antibiotic-resistant phenotypes in a variety of microbial species that naturally fill this niche. Antibiotic resistance phenotypes may spread horizontally to human, animal, or zoonotic diseases if they arise on a mobile genetic element. This might be very dangerous for public health. There is ample documentation of this environmental transmission pathway. For instance, environmental Gram-negative bacteria *Kluyvera* spp. are the source of the *bla*CTX-M genes, a clinically significant cause of antibiotic treatment failures [47]. Furthermore, the *Shewanellaceae* family of marine bacteria is the source of the *OXA*-48-type carbapenem-hydrolyzing β-lactamase gene, which is a significant contributor to the failure of antimicrobial treatments [46].

## 5. Antimicrobial Resistance and Animal Breeding

Antibiotic resistance has been linked to the use of antibiotics in agriculture, animal care, and human health. The creation and spread of bacteria with antibiotic-resistance characteristics are thought to be mostly caused by the industrial usage of antibiotics for the purpose of promoting animal growth. First, in the United Kingdom, in the 1960s, a link was identified between the use of antibiotics as growth promoters and a rise in multidrug resistance. The Swann Commission made this observation and also suggested that the use of antibiotics relevant to humans as growth enhancers for food animals be outlawed [101]. A significant advancement in the control of antibiotic use in the agricultural sector was achieved by the European Union in 1999 when it outlawed the use of four classes of antibiotics as growth promoters. This was followed by the outlawing of all growth promotion classes of antibiotics in animals that produced food in 2006 [102]. The introduction of these laws has prompted discussions over whether the new policies, which aim to drastically cut the amount of antibiotics used in animals raised for food, would have any impact on the development and spread of antibiotic resistance, especially in the human population. To ascertain the relationships between policies that restricted or outlawed the use of antibiotics in animals raised for food and the incidence of antibiotic resistance in both people and animals, Tang et al. [103] conducted a first systematic review and a meta-analysis. To find studies that sought to establish a connection between any actions taken to lessen the use of antibiotics in animals raised for food and the incidence of antibiotic resistance in both people and animals, this study team explored a wide range of electronic databases and grey literature. This research group discovered an association between the use of antibiotics in animals used for food production and the emergence of antibiotic resistance in the animals under study. This conclusion was based on 179 and 21 studies that reported the results of antibiotic resistance in animals and humans, respectively. They also discovered a comparable relationship between a decline in the use of antibiotics in food animals and a decline in the prevalence of antibiotic resistance in people, particularly in those who had close contact with animals that produced food. However, the reduction in antibiotic use is not unhampered by negative effects. Using the publicly accessible databases MEDLINE and AGRICOLA, McEwen et al. [104] conducted a keyword search to find publications that discussed unexpected effects related to national limits on the use of antibiotics in animals raised for food. Only a small number of studies that were solely from Europe were found via the search. Weanling pigs saw early increases in diarrhea when Sweden and Denmark banned the use of antibiotic growth promoters (AGPs); in other food-animal species, there was either no rise in diarrhea or just a little increase [105]. Improvements in animal housing, cleanliness, and health management were determined to be the primary factors in the effective resolution of these early issues with diarrhea episodes in both nations [106]. Preventive measures can limit the “inconveniences” of reducing the use of antimicrobials. Housing circumstances (production type) have an impact on the prevalence of resistant fecal coliforms in swine, according to Langlois et al. (1988) [107]. Pigs in the finishing unit have greater levels of resistant coliforms than pigs in the farrowing house or on pasture. Similar to this, fattening calves have a greater level of resistance to respiratory tract Pasteurellaceae [108] and fecal coliforms [109] than dairy calves. Additionally, Walson et al. (2001) [110] found that when population density dropped, the prevalence of antibiotic resistance was considerably reduced. Therefore, good farm management also decreases the cases in which antimicrobial drug use would be required.

## 6. Alternatives to Antibiotics

For the most part, the spectrum of viable substitutes for antimicrobials in farm animals is similar to that of human medicine [111]. Prebiotics and probiotics are widely accessible at the moment; however, it is uncertain, and probably varies, how effective they are. Combining the two has also been suggested; these are known as “synbiotics”. Other options include the use of bacteriophages and bacteriocins [112]. Bacteriophages are viruses that replicate by using the cells of bacteria. Phage treatment has been shown to be successful in treating *Salmonella Typhimurium* in pigs and poultry, but it also needs a quick phage selection and delivery [111]. ListShieldTM and ListexTM P100 [113], two commercial *Listeria* phage products, are now authorized as food preservatives. The effectiveness of this novel product against *Listeria monocytogenes* has been investigated in many studies. For example, a 5-log decrease in *L. monocytogenes* was found by Soni and Nannapaneni [114] after a 24 h treatment with ListexTM P100 at room temperature. This product was also tested against *L. monocytogenes* biofilms that had already formed. Iacumin et al. [115] noted *Listeria monocytogenes* biofilm’s total disintegration on wafers made of stainless steel after a 24 h application of ListexTM P100 at 20 °C. Only a 2-log drop in *Listeria* was shown by other authors during a shorter treatment period (two hours) [116]. The utilization of bacteriocins, a class of ribosome-produced proteins or peptides with antibacterial characteristics, is a possible substitute for antibiotics [117]. Because of their unique mechanism of action, bacteriocins have garnered a lot of interest in the field of antimicrobial research during the last 10 years. These antimicrobial peptides or proteins may have a bacteriostatic (i.e., preventing cell growth) or bactericidal (i.e., inducing cell death) impact, depending on the type of bacteriocins. Certain kinds of bacteriocins target lipid II, an integrative molecule of the bacterial cell membrane, to block the formation of peptidoglycans [118]. A pore is created in the bacterial cell membrane by other kinds of bacteriocins that have a binding affinity for lipid II. This results in a decrease in turgor, an interruption of the electrochemical gradient, and, ultimately, cell death [119]. Different groups of bacteriocins have the ability to block core metabolic activities such as protein synthesis, DNA replication, and gene expression, in addition to their targeting of the bacterial cell wall envelope [120]. While many other types of bacteria are capable of producing bacteriocins, lactic acid bacteria (LAB) have received the majority of attention due to their high production of these antimicrobial peptides. Three LAB species—*Lactococcus lactis*, *Lactobacillus sakei*, and *Lactobacillus curvatus*—were tested by Gomez et al. [121] against biofilms of important foodborne pathogens, *L. monocytogenes*, *E. coli*, and *S. typohimurium* serovar Enterica. A different study team [122] used nisin, a bacteriocin that is authorized for use in commerce, to combat *L. monocytogenes* biofilm over the course of nine hours, and they saw a 3.5-log decrease in 48 h. The variety of bacteriocins is rather high. It has been discovered that LAB produces more than 230 distinct bacteriocins, according to Alverz-Sieiro et al. [123]. These intriguing antimicrobial peptides have been explored to a limited extent. None of the above-mentioned options, however, are anywhere near to becoming globally commercially accessible to combat the whole range of microbial diseases in farm animals. Increased vaccination options for veterinary usage might be a more doable idea right now. Vaccines against bacterial infection and sickness are now not routinely used despite the fact that they are already available against many of the main viral diseases that affect cattle. A live oral Lawsonia vaccination study in pigs showed 80% decreased oxytetracycline consumption and enhanced productivity [124], but the vaccine is not commonly utilized. Using livestock that is genetically immune to diseases by the application of genetic modification technologies might be a longer-term strategy for lowering the use of antibiotics in farm animals. The creation of transgenic hens that are incapable of spreading avian influenza is one instance of early achievement in this area [125]. In addition to the options listed above, there has been encouraging progress in employing molecule combinations to take advantage of the collateral susceptibility of bacteria that would otherwise be resistant to antibiotics. When combined with clavulanic acid, a β-lactamase inhibitor, a considerable percentage of MRSA isolates, including the pandemic USA300 lineage, may convert to penicillin susceptibility, as shown by Harrison et al. (2019) [126].

## 7. Actions to Counteract Antimicrobial Resistance

The problems posed by the establishment and spread of antibiotic resistance linked to livestock antibiotic use are very complicated and affect not only people but also animals in many different ways. Crucially, the most current projections show that the world’s antimicrobial usage in food-producing animals would rise by 67% between 2010 and 2030, from 63,151 ± 1560 tons to 105,596 tons [127]. It is estimated that changing production methods in middle-income nations’ cattle sectors will account for one-third of this worldwide rise [127]. In fact, China and India represent the largest hotspots of resistance, with new hotspots emerging in Brazil and Kenya. This is according to the most recent report written by Van Boeckel et al. (2019) [128] regarding antimicrobial resistance of pathogens isolated from animals [128]. The same research group suggested that high-income nations, where the use of antibiotics on farms dates back to the 1950s, should aid low- and middle-income nations in their shift to sustainable animal husbandry [128]. Currently, 175,000 tons of antibiotics are produced, used, and misused annually by humans [129]. However, without a doubt, current antibiotic consumption cannot be sustained in the long term, and many researchers believe that, at this rate, the establishment of antimicrobial AMR is inevitable. For a drastic reduction in antibiotic use, it would be imperative to provide technical and financial support to developing countries to implement the WHO Global Action Plan and the recommendations of the Food and Agriculture Organization (FAO) [130]. Global implementation of the WHO’s recommended strategies is critical in the battle against the establishment and spread of antibiotic resistance. A number of organizations collaborate within the European Union’s borders to address the emergency that is represented by AMR, such as the “European Centre for Disease Prevention and Control”, the “European Committee on Antimicrobial Susceptibility Testing”, the “European Food Safety Authority”, and the “European Medicines Agency”. Using a surveillance network, the European Center for Disease Prevention and Control (ECDC) is an independent agency of the European Union that was established in 2004 with the goal of enhancing and fortifying member nations’ defenses against infectious illnesses. Specifically, it helps to preserve the efficacy of antibiotics by giving European countries a thorough and up-to-date annual analysis of the antibiotic resistance issue and encouraging the responsible use of these medications. The ECDC oversees a number of tasks, including surveillance, which enables the early detection of hazards to the public’s health, as well as validation, evaluation, and investigation of those risks in order to provide a framework for the state-by-state implementation of internal control systems. Established in 1997, the European group on Antimicrobial Susceptibility Testing (EUCAST) is a permanent group that is coordinated by the European national breakpoint committees. This takes care of defining the technical parameters and breakpoints for in vitro antimicrobial susceptibility testing, standardizing the data that emerges from the common techniques employed by different Member States. EUCAST, in particular, released “the lines EUCAST guide for the identification of resistance mechanisms and resistances specifications of clinical and/or epidemiological importance” in 2017. This document seems to be quite significant for European surveillance operations. In the fight against antimicrobial resistance (AMR), the European Food Safety Authority (EFSA), the European Medicines Agency (EMA), and the European Centre for Disease Prevention (ECD) are devoted to helping parties manage the causes and effects of this emergency in the food chain and in animals by offering scientific and educational support and consultancy services. In an effort to present a comprehensive picture of the AMR situation in Europe, EFSA gathers and disseminates data on zoonotic diseases across national borders. Based on the data obtained, EFSA and ECDC collaborate to produce annual summary reports regarding outbreaks of origin food, infections, and relevant resistance. The annual reports are reviewed by EFSA’s scientific experts, who then provide recommendations for reducing and preventing the use of antibiotics. These organizations should collaborate to write a program of intent to be pursued in the near future to reduce the use of antimicrobial drugs. In Europe, a small but important step toward a more rational use of antimicrobial molecules can be seen in the enactment of Regulation (EU) 2019/4. Regulation (EU) 2019/4 mediates the manufacture, market, and use of medicated feed. The regulation aims to achieve a high level of human health protection, high quality and safety standards for production, and increased availability of medicated feed. In general, the Regulation contributes to the Union’s action to combat antimicrobial resistance: it bans the use of antibiotics in medicated feed as prophylaxis or growth promoters, limits the validity and duration of veterinary prescriptions for antimicrobials, and establishes harmonized limits for antimicrobial residues. It thus promotes a more responsible use of antimicrobials in combating antimicrobial resistance in animals and preventing the spread of antibiotic-resistant bacteria through the food chain. The regulation, which repeals Directive 90/167/EEC, sets standards for each stage from production to distribution and the conditions for the use of medicated feed. Medicated feeds are to be used on veterinary prescription and for animals for which the veterinary prescription has been issued.

## 8. Conclusions

Governments, international organizations, and scientists must work together to respond to AMR effectively. Research contributions from a variety of fields are clearly needed, including social science, international law, health economics, epidemiology, microbiology, and pharmacology, in addition to clinical and veterinary medicine. Promptly updated data and effective strategies to combat AMR must be provided to human doctors, veterinary doctors, and pharmacists, who will have to modulate therapeutic choices in the fight against the dangerous phenomenon of AMR. Clinicians, pharmacists, patients, veterinarians, and farmers will need to be highly educated to establish long-term solutions to the problem. In the immediate future, however, there is a need for the introduction of a revitalized antimicrobial drug pipeline and viable substitutes that can be used long-term for the treatment of microbial diseases in livestock and humans.

## Figures and Tables

**Figure 1 antibiotics-13-00551-f001:**
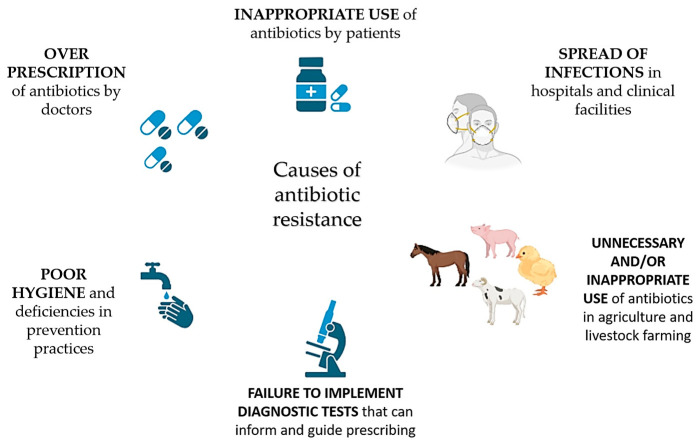
Main causes of antibiotic resistance.

**Figure 2 antibiotics-13-00551-f002:**
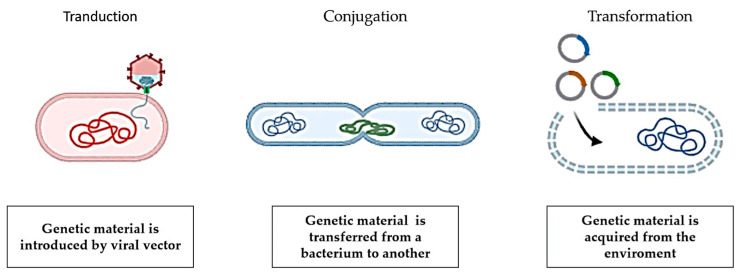
Horizontal gene transfer in bacteria.

**Table 1 antibiotics-13-00551-t001:** Mechanisms of antimicrobial resistance.

Modification of cell permeability	reduction of entry channels; efflux pumps
Production of inactivating enzymes	β-lactamase; acetyltransferase; phosphotransferase; adenyltransferase
Modification of the attachment site	penicillin-binding proteins (PBP); RNA polymerase
Activation of alternative metabolic pathway	modified enzymes

**Table 2 antibiotics-13-00551-t002:** Bacteria typical of antimicrobial resistance in livestock and resistance-related genes.

Bacteria	ARGs
Gram positive: *Acinetobacter*, *Aeromonas*, *Clostridium*, *E. cloacae*, *E. coli*, *K. pneumoniae*, *P. aeruginosa, Salmonella*, *Sphingomonas*, *Vibrio*, etc.	β-Lactams: *bla*_CTX−M−1_, *bla*_CTX−M−8_, *bla*_CTX−M−14_, *mec*A, *amp*C, etc. Aminoglycosides: *aac*, *aad*, etc.
Gram negative: *E. faecium*, *E. faecalis*, *E. hirae*, *E. durans*, *E. caaeliflavus*, *E. avium*, *S. agalactiae*, *S. aureus*, *S. intermedius*, *S. hyicus*, etc.	Tetracyclines: *tet*A, *tet*B, *tet*C, *tet*G, *tet*O, etc. Sulfonamides: *sul*I, *sul*II, *sul*3, etc. MLSB: *erm*A, *erm*B, etc. Vancomycin: *van* Colistin: *mcr*-1 and *mcr*-5.1

## Data Availability

Not applicable.

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
