# Peer review of "Antimicrobial Resistance in Livestock: A Serious Threat to Public Health"

_antibiotics, 2024, doi:10.3390/antibiotics13060551_

Round 1
Reviewer 1 Report
Comments and Suggestions for Authors
The manuscript addresses the implications of AMR in public health. However, it appears disorganized and lacks clarity in its presentation. One key term is "One Health approach," but it is not addressed in the introduction and should be addressed by the authors.
Abstract: It is suggested that this section be reviewed. For instance, the first sentence is unclear and should begin with "AMR." Lines 26-29: Not only in hospital settings but also in patients, where the problem of AMR arises due to ineffective treatment of systemic infections. Lines 29-31: Only carbapenems? You may mention the presence of beta-lactamases (ESBL). Lines 29-40: This should be moved to line 23. Line 45: There should be a general introductory paragraph on the relevance of the topic. For example, referring to lines 52-55. Lines 56-120: This paragraph is too long and loses clarity due to its length. Specific aspects should be addressed separately, such as including Figure 1 at line 59 to contextualize the information. Lines 64-121: It is suggested to separate causes, food production, food as a transmission route, microorganisms, resistance genes, among others. What is the objective of this review? Clarifying this would lead to a smoother reading experience. Line 125: An explanatory table is insufficient; it is suggested to add a figure showing the main mechanisms of AMR. Lines 142-248: Gene names should be in lowercase and italicized, protein names with the initial letter capitalized and not italicized. Bacterial species names should be italicized, e.g., line 244. The text should be reorganized as it is confusing in some sections. Why are important genes like tetA, clpC, fep, mcr not mentioned? It is surprising that Listeria monocytogenes is not included, considering livestock. Therefore, it would be clarifying to know the main microorganisms associated with MDR in livestock. Lines 279-329: This should come first, followed by lines 331-378. I believe causes should be addressed first, followed by the effects of antibiotic misuse. Logical sequence: Causes of MDR, effects, consequences, and then control measures or strategies. Line 380: This section is well placed, but the preceding sections need to be rearranged. Line 444: How does this align with the objective of this review? Line 512: How does the conclusion align with the study's objective? Is it a problem or not? Lines 513-515: Why is this a conclusion, and how does it relate to the findings? Lines 519-521: What strategies? The conclusions need to be rewritten.
Author Response
Reviewer 1
The manuscript addresses the implications of AMR in public health. However, it appears disorganized and lacks clarity in its presentation.
R: the authors all thank you deeply for your important revision work. The manuscript has been completely revised and reworded based on your suggestions.
One key term is "One Health approach," but it is not addressed in the introduction and should be addressed by the authors.
Response: many thanks for this comment, the term “One-health” now appears in the introductory section
Abstract: It is suggested that this section be reviewed. For instance, the first sentence is unclear and should begin with "AMR."
R: many thanks for your comment, the abstract has been revised and the initial sentence has been reworded as indicated
Lines 26-29: Not only in hospital settings but also in patients, where the problem of AMR arises due to ineffective treatment of systemic infections.
R: as per your suggestion, immediately after the sentence regarding the problem of antibiotic resistance in hospital care, a sentence regarding the indiscriminate consumption of antibiotics by patients was added.
Lines 29-31: Only carbapenems? You may mention the presence of beta-lactamases (ESBL). Lines 29-40: This should be moved to line 23.
R: as per your suggestion a sentence about ESBL bacteria has been inserted.
Line 45: There should be a general introductory paragraph on the relevance of the topic. For example, referring to lines 52-55.
R: thanks for your important suggestion an introductory paragraph to the following section has been added
Lines 56-120: This paragraph is too long and loses clarity due to its length. Specific aspects should be addressed separately, such as including Figure 1 at line 59 to contextualize the information.
R: we have rewritten the introduction also in relation to the indications of the other reviewers. The reference to figure 1 was inserted immediately after the sentence you indicated.
Lines 64-121: It is suggested to separate causes, food production, food as a transmission route, microorganisms, resistance genes, among others. What is the objective of this review? Clarifying this would lead to a smoother reading experience.
R: the introduction section has been rewritten; the objective of the review was better defined in the final part of the introduction of the manuscript
Line 125: An explanatory table is insufficient; it is suggested to add a figure showing the main mechanisms of AMR.
R: as per your valuable advice, for which we thank you, we have added a table and a figure.
Lines 142-248: Gene names should be in lowercase and italicized, protein names with the initial letter capitalized and not italicized.
R: the text has been checked entirely to remedy these formatting errors when present
Bacterial species names should be italicized, e.g., line 244.
R: the text has been checked entirely to remedy these formatting errors when present
The text should be reorganized as it is confusing in some sections. Why are important genes like tetA, clpC, fep, mcr not mentioned? It is surprising that Listeria monocytogenes is not included, considering livestock. Therefore, it would be clarifying to know the main microorganisms associated with MDR in livestock.
R: we have inserted sentences about the genes you recommended into the manuscript. The main bacteria that have demonstrated antimicrobial resistance have been summarized in a table
Lines 279-329: This should come first, followed by lines 331-378. I believe causes should be addressed first, followed by the effects of antibiotic misuse. Logical sequence: Causes of MDR, effects, consequences, and then control measures or strategies.
R: we have moved the paragraph as indicated
Line 380: This section is well placed, but the preceding sections need to be rearranged. Line 444: How does this align with the objective of this review?
R: thanks for tour comment, all sections have been profoundly modified and the title of the indicated one has also been modified
Line 512: How does the conclusion align with the study's objective? Is it a problem or not?
R: this section has been modified and, as indicated in line with the final part of the introduction which states the objectives of the manuscript, solutions to combat antibiotic resistance have been proposed.
Lines 513-515: Why is this a conclusion, and how does it relate to the findings?
R: title of the indicated section has been modified and the section has been improved in its content
Lines 519-521: What strategies? The conclusions need to be rewritten.
R: the conclusion has been rewritten

Reviewer 2 Report
Comments and Suggestions for Authors
This review article aims to discuss the issues of AMR in livestock and link this to public health. Although the text has few relevant contents the article is unorganized and scattered contents. No crisp figures and tables for the readers. No precise data on how relevant the AMR issue is (especially the transfer from livestock to humans).
Line 33: What’s the message here? AMR strains can clear up on their own so why to worry about AMRs? regarding biofilm related issues this article may be referrred: https://doi.org/10.1016/j.tim.2020.03.014
Line 38-40: rewrite the sentence.
Lines 307-318 are not linked to each other (missing continuity).
Line 341-344: incomplete contents with abrupt ending.
Lines 457-459: rewrite
Line 448-49 and line 471 values of worldwide usage do not corroborate. Section 5 title needs revision. In addition, the strategies mentioned may be supported by a few clinical trials or industrial products with reference.
The conclusion section is weak. Avoid blaming anyone (patients, veterinarians and farmers) for this.
The similarity of the text material is more than 25%, authors maybe advised to reduced the similarity where the whole sentences are taken from elsewhere.
Comments on the Quality of English LanguageSo many redundant and non-continuity between sentences. Grammatical errors, sentence structure and word choice needs to be corrected.
Author Response
This review article aims to discuss the issues of AMR in livestock and link this to public health. Although the text has few relevant contents the article is unorganized and scattered contents. No crisp figures and tables for the readers. No precise data on how relevant the AMR issue is (especially the transfer from livestock to humans).
Response: the authors all thank you deeply for your important revision work. The manuscript has been completely revised and reworded based on your suggestions.
Line 33: What’s the message here? AMR strains can clear up on their own so why to worry about AMRs? regarding biofilm related issues this article may be referrred: https://doi.org/10.1016/j.tim.2020.03.014
R: the sentences have been deleted
Line 38-40: rewrite the sentence.
R: the sentence has been rewritten
Lines 307-318 are not linked to each other (missing continuity).
R: the text has been modified and a link has been provided
Line 341-344: incomplete contents with abrupt ending.
R: the sentence has been rewritten
Lines 457-459: rewrite
R: the sentence has been rewritten
Line 448-49 and line 471 values of worldwide usage do not corroborate. Section 5 title needs revision. In addition, the strategies mentioned may be supported by a few clinical trials or industrial products with reference.
R: it was specified that the first data relate to the consumption of antibiotics in food-producing animals. The title of the section 5 has been modified. Clinical studies have been provided at support
The conclusion section is weak. Avoid blaming anyone (patients, veterinarians and farmers) for this.
R: this section has been rewritten
The similarity of the text material is more than 25%, authors maybe advised to reduced the similarity where the whole sentences are taken from elsewhere.
R: similarity has been diminished
So many redundant and non-continuity between sentences. Grammatical errors, sentence structure and word choice needs to be corrected.
R: the text has been deeply modified and checked according to the suggestion

Reviewer 3 Report
Comments and Suggestions for Authors
The review article „Antimicrobial resistance in livestock: a serious threat to public health” is a comprehensive review, well-written for the most part. The article discusses the impact of antibiotic use in livestock on the development of antibiotic-resistant strains. It also discusses the strategies and current efforts to reduce antibiotic use in livestock and the efforts to prevent the spread of antibiotic resistance. The comments are as follows:
1. The section titled The Mechanisms of Antimicrobial Resistance is too long, so I suggest shortening it. It would also be helpful to discuss the mechanisms of resistance at the beginning of the section rather than at the end. The classification of efflux pump families should be mentioned before the discussion of efflux pump-related mutations in Line 157.
Line 125: The sentence is too long, I suggest splitting the sentence to make it easier to understand.
Line 145: The sentence makes no sense.
Line 171: The mechanisms of horizontal transmission and horizontal gene transfer are discussed in line 171 and repeated again in the Line 180
Line 195: Define extended spectrum β-lactamases
Author Response
The review article “Antimicrobial resistance in livestock: a serious threat to public health” is a comprehensive review, well-written for the most part. The article discusses the impact of antibiotic use in livestock on the development of antibiotic-resistant strains. It also discusses the strategies and current efforts to reduce antibiotic use in livestock and the efforts to prevent the spread of antibiotic resistance. The comments are as follows:
Response: many thanks for your revision work. We have modified the manuscript as per your suggestions
- The section titled The Mechanisms of Antimicrobial Resistance is too long, so I suggest shortening it. It would also be helpful to discuss the mechanisms of resistance at the beginning of the section rather than at the end. The classification of efflux pump families should be mentioned before the discussion of efflux pump-related mutations in Line 157.
R: the section has been rewritten and divided into two parts. An additional paragraph has in fact been created. The first paragraph describes the genetic factors underlying the transmission of drug resistance; in the second section we describe the mechanisms that the bacterium expresses to resist the action of the drug.
Line 125: The sentence is too long, I suggest splitting the sentence to make it easier to understand.
R: the sentence has been modified
Line 145: The sentence makes no sense.
R: the sentence has been rewritten
Line 171: The mechanisms of horizontal transmission and horizontal gene transfer are discussed in line 171 and repeated again in the Line 180
R: the redundant sentence has been deleted
Line 195: Define extended spectrum β-lactamases
R: an explanation has been provided
Round 2
Reviewer 1 Report
Comments and Suggestions for Authors
L33: under the acronym ESBL (Extended-spectrum Betalactamase change by: Extended-spectrum Betalactamase (ESBL)…..
L380: generate plasmid-encoded EreA and EreB esterases is incorrcet. Replace by generate plasmid-encoded ereA and ereB esterases genes
L477-L480-L483-L492: coli not Coli
L491: E. coli in italics
L517: add gene after resistant
L536: in italics blaCTX-M gene
L536: blaCTX-M in italics
L537: Shewanellaceae in italics
L538: in italics, it is gene
L351 (point 3) to L539: it is necessary to review these new paragraphs since there are omissions regarding the nomenclature of genes and proteins.
Author Response
Reviewer 1
Response: many thanks for your revision work. We have revised the manuscript according to your suggestions
L33: under the acronym ESBL (Extended-spectrum Betalactamase change by: Extended-spectrum Betalactamase (ESBL)…..
R: now amended
L380: generate plasmid-encoded EreA and EreB esterases is incorrcet. Replace by generate plasmid-encoded ereA and ereB esterases genes
R: replaced
L477-L480-L483-L492: coli not Coli
R: the error has been corrected in the text
L491: E. coli in italics
R: now amended
L517: add gene after resistant
R: now amended
L536: in italics blaCTX-M gene
R: now amended
L536: blaCTX-M in italics
R: now amended
L537: Shewanellaceae in italics
R: now amended
L538: in italics, it is gene
R: now amended
L351 (point 3) to L539: it is necessary to review these new paragraphs since there are omissions regarding the nomenclature of genes and proteins.
R: the paragraph has been amended accordingly
